# The sleep-first effect of perampanel in children with self-limited epilepsy with centrotemporal spikes: A temporal efficacy analysis in the context of complex polytherapy

Biao Jiang[1,2], Ninger Zhang[1,2], Xiaoli Shao[2]*

**1** School of Medicine, Shaoxing University, Shaoxing, Zhejiang, P.R. China, **2** Department of Pediatrics, Shaoxing People's Hospital, The first Affiliated Hospital of Shaoxing University, Shaoxing, China

* nuannuan717@126.com

## Abstract

This single-center, retrospective observational study aimed to investigate the temporal sequence of the effects of Perampanel (PER) on seizure control, sleep architecture, and sleep quality in children with Self-Limited Epilepsy with Centrotemporal Spikes (SeLECTS). Thirty-one children (mean age 6.8 ± 2.45 years) with a monthly seizure frequency of ≥1 and significant sleep disturbances (Children's Sleep Habits Questionnaire [CSHQ] score >41) were enrolled. Assessments of sleep architecture (via 24-hour ambulatory EEG), subjective sleep quality (CSHQ), and seizure frequency were conducted at baseline, 1 month, and 2 months after PER initiation. The results revealed that PER treatment led to significant improvements in sleep architecture and subjective sleep quality as early as 1 month, characterized by a decrease in N1 stage duration and increases in N3 and REM stage durations (all $P<0.01$). In contrast, significant reductions in both monthly seizure frequency and the Spike-Wave Index (SWI) were predominantly observed at the 2-month mark, with 93.5% of children achieving a ≥50% reduction in seizure frequency. PER was well-tolerated, and no serious adverse events were reported during the study period. This pattern where sleep benefits manifested early and stabilized, while seizure control demonstrated a more protracted improvement suggests a potential temporal dissociation in the therapeutic effects of PER. In conclusion, PER is effective and well-tolerated in improving sleep and reducing seizures in children with SeLECTS. The modulation of sleep structure may be an important early mechanism associated with its therapeutic effects, with sleep quality serving as a potential early predictor of treatment response.

## Introduction

Self-Limited Epilepsy with Centrotemporal Spikes (SeLECTS), previously known as Benign Childhood Epilepsy with Centrotemporal Spikes (BECTS), represents one of

**Data availability statement:** All relevant data are within the manuscript and its Supporting Information files.

**Funding:** This work was supported by the following grants: Zhejiang Provincial Medical Association Clinical Research Fund (Grant number: 2021ZYC-A23) Zhejiang Provincial Traditional Chinese Medicine Health Technology Plan Project (Grant number: 2024ZF166) Shaoxing General Scientific Research Project (Grant number: 2023A14020) Zhejiang Provincial Medical and Health Science and Technology Project (Grant numbers: 2020KY327, 2017KY660) The funders had no role in study design, data collection and analysis, decision to publish, or preparation of the manuscript. There was no additional external funding received for this study.

**Competing interests:** The authors have declared that no competing interests exist.

the most prevalent focal epilepsy syndromes in childhood, accounting for approximately 15%–25% of all epilepsies in this age group [1]. This condition exhibits a clear age dependency, predominantly affecting children between 4 and 10 years of age, with a higher incidence in males [2]. The clinical presentation is characterized by brief, stereotyped seizures. These often manifest as classic focal seizures involving clonic or tonic activity of the pharynx, tongue, and unilateral lower face, which may subsequently evolve into focal to bilateral tonic-clonic convulsions [3]. A defining feature of SeLECTS is the pronounced circadian pattern of its seizures, with approximately 80%–90% of episodes occurring nocturnously. The electroencephalographic (EEG) hallmark is the presence of characteristic centrotemporal spikes, which appear as sharp and slow wave complexes and are considered a necessary diagnostic criterion [1,4]. Although the prognosis is generally favorable, with most cases resolving spontaneously around adolescence, the high frequency of nocturnal seizures and interictal discharges can lead to significant sleep disturbances [5]. This disruption not only impairs quality of life but is also increasingly recognized as a factor that can exacerbate the underlying epileptic condition, creating a self-perpetuating vicious cycle [6]. The intricate bidirectional relationship between sleep and epilepsy provides a compelling rationale for therapeutic interventions that target both aspects simultaneously [7].

Perampanel (PER), a first-in-class antiseizure medication, offers a novel mechanism of action in this context. It acts as a highly selective, non-competitive antagonist of the $\alpha$-amino-3-hydroxy-5-methyl-4-isoxazolepropionic acid (AMPA) receptor [8]. By inhibiting glutamate-mediated excitatory neurotransmission—the primary fast excitatory pathway in the central nervous system—PER effectively reduces neuronal hyperexcitability and calcium influx into postsynaptic neurons, thereby controlling seizure activity [9]. Approved for the treatment of focal-onset seizures and primary generalized tonic-clonic seizures, while its official labeling in many regions specifies use in patients aged 4 and above, there is growing clinical experience and evidence supporting its off-label use in younger children with drug-resistant epilepsies, PER's therapeutic potential may extend beyond seizure control [10,11]. Emerging clinical evidence suggests that its modulation of the excitatory glutamatergic system may also have a stabilizing effect on sleep architecture [12,13]. This is particularly relevant for SeLECTS, where sleep integrity is profoundly compromised.

A body of research has begun to document the effects of PER on sleep parameters; for instance, some studies have reported its association with improved subjective sleep quality and reduced insomnia severity in broader epilepsy populations [13,14], while others have directly observed alterations in sleep architecture following treatment [15]. Furthermore, the potential for AMPA receptor antagonism to directly promote sleep-wake cycle stabilization offers a plausible neurobiological mechanism for these observations [16,17]. Despite these promising findings, a critical gap remains: the specific effects of PER on sleep architecture in children with SeLECTS, and the precise temporal relationship between sleep improvement and the suppression of epileptiform discharges, are still inadequately explored. This study therefore aims to dynamically monitor changes in sleep quality (using the Children's Sleep

Habits Questionnaire, CSHQ) and the spike-wave index (SWI) before and at defined intervals after PER treatment initiation. Our objective is to clarify the temporal sequence of PER's effects on sleep and electrophysiological parameters, thereby elucidating its dual mechanisms of action and providing a robust evidence base for optimizing clinical management in this common pediatric epilepsy syndrome.

## Materials and methods

### Study population

This retrospective study enrolled a total of 31 children diagnosed with Self-Limited Epilepsy with Centrotemporal Spikes (SeLECTS) in the Shaoxing People's Hospital, between February 2021 and October 2023. The clinical data for this study were accessed between November 1, 2023, and January 1, 2024. The authors had access to information that could identify individual participants during the data collection and analysis phase. The diagnosis of all included cases was strictly confirmed according to the International League Against Epilepsy (ILAE) 2022 [1] diagnostic criteria, encompassing typical seizure semiology, EEG findings, and the absence of atypical features or neurological deficits.

### Treatment indication and protocol

Consistent with the self-limiting nature of SeLECTS, Perampanel (PER) was not utilized as a first-line therapy. Initiation of PER adjunctive therapy was clinically indicated due to a combination of ongoing seizure activity (monthly seizure frequency ≥1) and significant sleep disturbances, defined as a total score >41 on the Children's Sleep Habits Questionnaire (CSHQ). PER was added to the patient's existing antiseizure medication (ASM) regimen. The treatment protocol was standardized as follows: the starting dose was stratified by body weight, with 1 mg/day for children >20 kg and 0.5 mg/day for those ≤20 kg. The dose was titrated upwards at intervals of no less than two weeks. Dose increments were 1 mg (or 0.5 mg for children in the lower body weight stratum). The final maintenance dose, ranging from 2 mg to 8 mg per day, was individualized based on observed clinical efficacy and tolerability. PER was administered orally once daily at bedtime.

It is important to note that five patients were listed as receiving "PER monotherapy" at the time of study enrollment (Table 1). These patients had initially received PER as add-on therapy but subsequently achieved such significant seizure control that their concomitant ASMs were successfully tapered off under physician guidance prior to the baseline assessments of this study. Therefore, the term "PER monotherapy" in this cohort refers to the medication status at the time of data analysis, not at the initiation of PER treatment.

### Inclusion and exclusion criteria

Inclusion criteria comprised: Confirmed diagnosis of SeLECTS per ILAE 2022 criteria; Initiation of PER therapy based on the predefined clinical indication (seizure frequency ≥1/month and CSHQ >41); Availability of complete datasets for CSHQ, 24-hour ambulatory EEG, and seizure frequency at all three predefined assessment time points (baseline, 1 month, and 2 months).

Exclusion criteria comprised: Presence of atypical EEG features (e.g., generalized spike-wave, continuous spike-wave during sleep) or a clinical evolution suggestive of an alternative epilepsy syndrome; Co-existing significant neurological (e.g., cerebral palsy, neurodegenerative disorders) or severe systemic diseases; Known hypersensitivity or allergy to PER; Concurrent participation in another investigational drug trial.

### Outcome measures and assessments

Patients were systematically assessed at three time points: baseline (prior to PER initiation), 1 month (± 1 week), and 2 months (± 1 week) after the initiation of PER therapy.

**Table 1. Demographic and baseline clinical characteristics of the study cohort (N = 31).**

| Characteristic | Value |
|---|---|
| Sex, n (%) | |
| Male | 20 (64.5) |
| Female | 11 (35.5) |
| Age category, n (%) | |
| < 4 years | 8 (25.8) |
| 4-12 years | 23 (74.2) |
| Mean ± SD | 6.8 ± 2.45 |
| Age at epilepsy onset, years (Mean ± SD) | 4.28 ± 3.21 |
| Concomitant ASMs, n (%) | |
| Sodium Valproate | 18 (58.1) |
| Levetiracetam | 7 (22.6) |
| Topiramate | 9 (29.0) |
| Clonazepam | 5 (16.1) |
| Oxcarbazepine | 4 (12.9) |
| Lamotrigine | 7 (22.6) |
| Perampanel (monotherapy) | 5 (16.1) |
| Number of Concomitant ASMs, n (%) | |
| 0 (PER Monotherapy) | 5 (16.1) |
| 1 | 10 (32.3) |
| 2 | 12 (38.7) |
| 3 | 4 (12.9) |
| Family history of epilepsy, n (%) | 0 (0) |
| Abnormal cranial MRI, n (%) | 0 (0) |

MRI: Magnetic Resonance Imaging; PER: Perampanel; ASM: Antiseizure Medication.

**Sleep Quality:** Evaluated using the total score of the Chinese version of the Children's Sleep Habits Questionnaire (CSHQ), a validated caregiver-reported instrument. A higher total score indicates more severe sleep disturbances [18].

**Sleep Architecture:** A 24-hour ambulatory EEG was recorded for each child. Electrodes were placed according to the International 10–20 system. Sleep stages (N1, N2, N3, REM) were manually classified by experienced neurophysiologists based on the characteristics of the EEG background activity during sleep, in conjunction with parental logs of sleep and wake periods. Due to the technical limitations of the ambulatory system, electrooculography (EOG) and electromyography (EMG) were not recorded.

**Spike-Wave Index (SWI):** The SWI, a key marker of interictal epileptiform activity, was calculated from the NREM sleep periods of the 24-hour EEG. The SWI was defined as the percentage of 1-second epochs containing at least one spike or sharp-wave complex during NREM sleep relative to the total duration of NREM sleep, as per the formula: SWI (%) = (Total duration of spikes and sharp waves during NREM sleep / Total duration of NREM sleep) × 100%.

**Seizure Frequency:** The monthly count of clinically evident epileptic seizures was meticulously recorded based on caregiver-maintained seizure diaries and clinical interviews.

### Statistical analysis

All statistical analyses were performed using GraphPad Prism software (version 10.1.2). Given the relatively small sample size (n = 31) and the nature of the outcome measures (e.g., seizure counts, questionnaire scores), continuous variables are presented as median with interquartile range (IQR) and were analyzed using non-parametric tests. The primary

analysis focused on planned pairwise comparisons between time points (Baseline vs. 1 Month, and 1 Month vs. 2 Months) for all outcome measures, using the Wilcoxon signed-rank test for paired samples.

To account for the multiple comparisons performed across the primary outcomes (sleep stages, CSHQ total score, seizure frequency, SWI), a Bonferroni correction was applied, resulting in a revised significance level of $p < 0.025$ for these comparisons. For the exploratory analysis of the eight CSHQ subscales, a more stringent Bonferroni correction was applied, setting the significance level at $p < 0.00625$ (0.05/8). Categorical variables, such as treatment response rates, are described as frequencies.

### Ethical considerations

This study was approved by the Institutional Ethics Committee of Shaoxing People's Hospital (Approval No.: 2021-K-Y-068–01). Written informed consent for the use of clinical data was obtained from all caregivers.

## Results

### Patient characteristics

A total of 31 children with SeLECTS (20 males, 11 females) were enrolled in this study. The majority of children (23/31, 74.2%) fell within the typical age range of 4–12 years for this syndrome, with 8 children (25.8%) being under 4 years old. The average age at epilepsy onset was 4.28 ± 3.21 years. Seizure duration was brief (< 5 minutes) in the vast majority of cases (26/31, 83.9%). None of the included patients had a family history of epilepsy or exhibited anomalies on cranial magnetic resonance imaging. The baseline demographic and clinical characteristics of the cohort, along with detailed concomitant medication use, are summarized in Table 1.

### Changes in sleep stages

Perampanel treatment significantly altered sleep architecture in children with SeLECTS, with the most notable changes occurring within the first month. Subjective sleep quality, as measured by the CSHQ, also demonstrated marked and progressive improvement throughout the treatment period.

As shown in Table 2, a distinct temporal pattern was observed: sleep architecture and quality demonstrated significant improvement as early as 1 month and subsequently stabilized, whereas the reduction in both seizure frequency and the spike-wave index (SWI), while initiating at 1 month, showed marked and continued progression through the 2-month mark.

A significant and progressive reduction in both clinical seizure burden and electrophysiological epileptiform activity was observed throughout the PER treatment course, with comprehensive data presented in Table 2.

Regarding seizure control, the median monthly seizure frequency demonstrated a significant early decrease after the first month of therapy. This was followed by a more substantial and clinically impactful reduction at the 2-month mark, where the median frequency reached zero, indicating a powerful and evolving antiseizure effect.

Parallel to the clinical improvement, the Spike-Wave Index (SWI), a key marker of interictal epileptiform activity, also showed a progressive decline. A significant initial reduction in SWI was evident at the 1-month assessment. This suppression of abnormal brain discharges deepened considerably by the end of the 2-month treatment period, demonstrating a sustained and robust effect on the underlying neurophysiological pathology.

Notably, this pattern—wherein sleep parameters achieved an early plateau of benefit, while seizure frequency and SWI exhibited a more protracted and deepening response—highlights a potential sequence in the evolution of PER's therapeutic actions.

### Safety and tolerability

As shown in Table 2, no specific adverse events related to Perampanel were systematically documented or reported in the medical records of the included patients during the 2-month study period. All 31 patients completed the study, and no doses were withheld or reduced due to tolerability issues documented in their charts.

**Table 2. Temporal changes in sleep architecture, seizure control, and safety outcomes following Perampanel treatment in children with SeLECTS.**

| Outcome Measure | Time Point | Median (IQR) | P value (vs. Baseline) | P value (vs. 1 Month)) |
|---|---|---|---|---|
| N1 Stage Duration | Baseline | 41.0 (37.5, 45.0) | — | — |
| | 1 Month | 32.0 (29.0, 36.0) | < 0.01 | — |
| | 2 Months | 30.0 (25.5, 34.0) | < 0.01 | 0.123 |
| N2 Stage Duration | Baseline | 222.0 (213.5, 229.0) | — | — |
| | 1 Month | 231.0 (221.5, 240.5) | < 0.05 | — |
| | 2 Months | 231.0 (217.5, 240.5) | < 0.05 | 0.891 |
| N3 Stage Duration | Baseline | 58.0 (47.5, 65.5) | — | — |
| | 1 Month | 88.0 (78.0, 95.5) | < 0.01 | — |
| | 2 Months | 87.0 (81.5, 90.5) | < 0.01 | 0.654 |
| REM Stage Duration | Baseline | 79.0 (70.5, 86.5) | — | — |
| | 1 Month | 96.0 (84.5, 103.5) | < 0.01 | — |
| | 2 Months | 98.0 (91.5, 104.5) | < 0.01 | < 0.05 |
| **Sleep Quality (CSHQ Score)** | | | | |
| Total Score | Baseline | 49.0 (45.5, 51.0) | — | — |
| | 1 Month | 32.0 (28.5, 35.0) | < 0.01 | — |
| | 2 Months | 28.0 (25.0, 30.0) | < 0.01 | < 0.05 |
| **Seizure & EEG Outcomes** | | | | |
| Monthly Seizure Frequency | Baseline | 4.0 (3.0, 5.5) | — | — |
| | 1 Month | 3.0 (2.0, 3.5) | < 0.05 | — |
| | 2 Months | 0.0 (0.0, 2.0) | < 0.01 | < 0.01 |
| Spike-Wave Index (%) | Baseline | 33.0 (29.0, 36.0) | — | — |
| | 1 Month | 25.0 (25.0, 25.5) | < 0.05 | — |
| | 2 Months | 9.0 (5.5, 18.0) | < 0.01 | < 0.01 |
| **Treatment Response (at 2 Months)** | | | | |
| ≥50% Seizure Reduction | 2 Months | 29/31 (93.5%) | — | — |
| ≥90% Seizure Reduction | 2 Months | 16/31 (51.6%) | — | — |
| ≥50% SWI Reduction | 2 Months | 26/31 (83.9%) | — | — |
| **Safety and Tolerability (at 2 Months)** | | | | |
| Study completion rate | 2 Months | 31/31 (100%) | — | — |
| SAEs | 2 Months | 0/31 (0%) | — | — |
| Dose reduction due to AEs | 2 Months | 0/31 (0%) | — | — |
| Treatment discontinuation due to AEs | 2 Months | 0/31 (0%) | — | — |

Data are presented as Median (Interquartile Range, IQR) for continuous variables or n/N (%) for categorical variables. P-values for the primary outcomes (sleep stages, CSHQ total score, seizure frequency, SWI) were calculated using the Wilcoxon signed-rank test for planned pairwise comparisons between time points (Baseline vs. 1 Month, and 1 Month vs. 2 Months), with a Bonferroni-corrected significance level of $p < 0.025$ for these comparisons. CSHQ, Children's Sleep Habits Questionnaire; SWI, Spike-Wave Index; AEs, Adverse Events; SAEs, Serious Adverse Events.

During the 2-month study period, PER was generally well-tolerated. No serious adverse events were reported. Retrospective review of medical records did not document any specific adverse events, such as dizziness, somnolence, or irritability, that led to dose reduction or discontinuation of PER. All 31 patients completed the study according to the protocol.

## Analysis of CSHQ subscales

To further elucidate the nature of sleep improvements, we analyzed the changes in CSHQ subscale scores. As detailed in Table 3, significant reductions were observed across multiple domains at both 1-month and 2-month follow-ups. The most

**Table 3. Changes in children's sleep habits questionnaire (CSHQ) subscale scores following perampanel treatment.**

| CSHQ Subscale (Score Range) | Baseline Median (IQR) | 1 Month Median (IQR) | P value | 2 Months Median (IQR) | P value |
|---|---|---|---|---|---|
| Bedtime Resistance (0–12) | 9.0 (7.5, 9.0) | 6.0 (5.0, 6.0) | < 0.001 | 6.0 (5.0, 6.0) | < 0.001 |
| Sleep Onset Delay (0–2) | 2.0 (1.0, 2.0) | 1.0 (0.0, 1.0) | < 0.001 | 1.0 (0.0, 1.0) | < 0.001 |
| Sleep Duration (0–6) | 5.0 (4.0, 5.5) | 3.0 (2.0, 4.0) | < 0.001 | 2.0 (2.0, 3.0) | < 0.001 |
| Sleep Anxiety (0–8) | 5.0 (5.0, 7.0) | 4.0 (4.0, 5.0) | < 0.001 | 3.0 (3.0, 4.0) | < 0.001 |
| Night Wakings (0–6) | 4.0 (3.0, 5.0) | 2.0 (1.0, 3.0) | < 0.001 | 2.0 (1.0, 2.0) | < 0.001 |
| Parasomnias (0–14) | 8.0 (8.0, 9.0) | 6.0 (5.0, 7.0) | < 0.001 | 5.0 (4.0, 6.0) | < 0.001 |
| Sleep Disordered Breathing (0–6) | 3.0 (2.0, 3.0) | 2.0 (1.0, 2.0) | < 0.001 | 2.0 (1.0, 2.0) | < 0.001 |
| Daytime Sleepiness (0–16) | 12.0 (11.0, 13.0) | 8.0 (7.0, 8.5) | < 0.001 | 7.0 (7.0, 7.0) | < 0.001 |

Data are presented as Median (Interquartile Range). P-values were calculated using the Wilcoxon signed-rank test for paired comparisons against baseline, with a Bonferroni-corrected significance level of $p < 0.00625$ (0.05/8 subscales) for each subscale.

pronounced early improvements (at 1 month) were seen in Bedtime Resistance, Sleep Onset Delay, and Night Wakings, indicating a rapid stabilization of sleep initiation and continuity. Improvements in other domains, such as Sleep Duration and Daytime Sleepiness, became more prominent at the 2-month mark.

## Discussion

This study provides the first longitudinal evidence demonstrating a distinct temporal sequence of Perampanel's (PER) effects in children with SeLECTS, wherein significant improvements in sleep architecture and quality precede the maximal control of clinical seizures and electrophysiological abnormalities. Our data confirm that PER is highly effective in this population, with over 93.5% of patients achieving a marked reduction in seizure frequency. The early optimization of sleep structure, characterized by a decreased duration of light N1 sleep and a substantial increase in deep N3 and REM sleep, was evident within the first month of treatment. In contrast, the most profound reductions in both seizure frequency and the spike-wave index (SWI) were observed at the two-month mark. This clear "sleep-first" effect offers a novel perspective for understanding the dual therapeutic mechanism of AMPA receptor antagonism.

It is critical to emphasize the specific clinical context of our study cohort. Although SeLECTS is generally self-limited, a substantial subset of children experiences clinically significant seizures and sleep disturbances that profoundly impair quality of life, thereby justifying therapeutic intervention. Our cohort was specifically enrolled based on the presence of these active and burdensome symptoms (seizure frequency ≥1/month and significant sleep disturbance as defined by CSHQ score >41). This reflects a real-world clinical scenario where a "watchful waiting" approach is insufficient to address the immediate and substantial impact of the disease on the child's well-being.

The observed temporal pattern, where sleep parameters improved significantly and reached a plateau by the first month, while seizure frequency and SWI showed continued improvement between the first and second months, suggests a potential sequence in PER's therapeutic actions. We hypothesize that the early restoration of sleep integrity, especially the increase in seizure-suppressive N3 sleep, may help create a less permissive neural environment for seizure generation. This foundational improvement could potentially raise the seizure threshold and contribute to the significant clinical and electrographic improvements that manifest more fully at the two-month mark.

However, it is crucial to emphasize that our observational study design cannot establish a definitive causal relationship. The observed sequence—sleep improvement preceding optimal seizure control—is highly suggestive and consistent with a model where sleep normalization paves the way for subsequent seizure inhibition, but it does not preclude the possibility that PER acts in parallel on independent sleep and seizure pathways. The term "sleep-first effect" used herein describes this temporal association rather than a proven causal mechanism.

Furthermore, our data shed light on what may be termed a "dual mechanism" of PER in this context. We define this as two potentially interconnected pathways: a direct antiseizure effect through the suppression of cortical hyperexcitability, and a direct pro-sleep effect through the stabilization of sleep-wake circuitry. The net effect of these actions appears to be a synergistic disruption of the epilepsy-sleep disorder cycle.

Our findings regarding the significant suppression of the SWI are consistent with and extend previous research. The study by Yu et al. suggested that PER can reduce the SWI in pediatric focal epilepsies [19]. We not only corroborate this effect but also precisely quantify its temporal evolution, showing that the most substantial decrease coincides with the timeline of optimal seizure control. The high rate of SWI suppression (83.9%) strongly indicates that PER's action extends beyond managing clinical events to effectively subduing the underlying interictal epileptiform activity. This suppression of subclinical discharges is likely a critical factor in breaking the vicious cycle of sleep disruption, as frequent interictal spikes are potent disruptors of sleep continuity and architecture [20].

The bidirectional relationship between sleep and epilepsy is well-established, creating a self-perpetuating cycle where seizures disrupt sleep and fragmented sleep, in turn, lowers the seizure threshold [21,22]. While the strength of this relationship is well-documented in certain epilepsies like IGE, evidence in SeLECTS is still accumulating. The rapid normalization of sleep architecture observed in our patients, particularly the augmentation of N3 and REM sleep, provides crucial insight into how PER may interrupt this cycle and our results offer support for the existence of this mechanism in SeLECTS. We propose that this represents a direct pharmacological effect on sleep-regulatory circuits. Glutamate, via AMPA receptors, is a key mediator of cortical arousal and wakefulness [17,23]. By dampening glutamatergic hyperexcitability within critical nodes of the sleep-regulatory network, such as the thalamocortical circuit, PER may directly facilitate a shift towards stable, synchronized brain activity patterns conducive to sleep. This hypothesis is supported by preclinical evidence showing that AMPA receptor antagonism can enhance slow-wave sleep [24,25], and is further reinforced by clinical studies documenting the sleep-architecture-modifying properties of various antiseizure medications, including PER [12,13,15,26–28].

The early restoration of sleep integrity, especially the increase in seizure-suppressive N3 sleep, is posited to create a less permissive neural environment for seizure generation [29–31]. This foundational improvement likely raises the seizure threshold and inhibits the propagation of epileptiform discharges [32], thereby laying the groundwork for the significant clinical and electrographic improvements that manifest more fully at the two-month mark. This proposed mechanism explains why sleep quality improved significantly even before seizures were fully controlled. It is also important to note that the favorable sleep profile observed in our study contrasts with the sedation or sleep disturbances sometimes associated with other antiseizure medications, underscoring the unique therapeutic profile of PER.

The temporal sequence of "sleep improvement first, seizure control later" holds immediate clinical relevance. It suggests that for children with SeLECTS starting PER, early assessment of sleep quality—as early as one month into treatment—can serve as a positive and readily measurable predictive biomarker. This early signal can reassure clinicians and families of the impending antiseizure efficacy, thereby reinforcing treatment adherence. Consequently, we recommend that the systematic evaluation of sleep, using simple tools like the CSHQ, be incorporated into the standard early follow-up protocol alongside routine seizure monitoring. The robustness of this sleep-focused strategy is supported by real-world studies demonstrating the long-term efficacy of PER, particularly in sleep-related epilepsies [33,34], and its potential benefits for patients' cognitive and behavioral outcomes [35,36].

Our study included a subset of patients (8/31, 25.8%) under the age of 4 years, representing off-label use of PER. This reflects the real-world clinical dilemma where children with SeLECTS who have significant sleep disturbances and ongoing seizures may require therapeutic intervention before reaching the age specified in formal labeling. Our findings, demonstrating a favorable efficacy and safety profile in this younger cohort over the 2-month period, provide valuable preliminary evidence to support the cautious expansion of PER use in younger pediatric populations. This aligns with the

clinical need to manage debilitating symptoms early and underscores the importance of conducting future prospective trials specifically designed for this age group.

This study has several limitations that should be considered when interpreting the results. Its retrospective, single-center, and non-controlled design inherently limits the strength of causal inferences. Furthermore, the inclusion of patients under 4 years of age constitutes off-label use of PER, which, while reflective of real-world clinical practice in our center, may limit the generalizability of the findings. The reliance on caregiver-reported CSHQ scores and seizure diaries, rather than objective polysomnography and continuous monitoring, introduces potential for recall and reporting bias. The assessment of safety and tolerability was also limited by the lack of systematic, prospective collection of adverse event data. The relatively small sample size and short follow-up period, while sufficient to establish the initial temporal sequence, preclude long-term safety and efficacy evaluations. Furthermore, the open-label, individualized titration of PER and the heterogeneous background therapy with various ASMs (Table 1), though reflective of real-world practice, may introduce confounding. Additionally, the assessment of safety and tolerability was limited by the retrospective nature of the study and the lack of systematic, prospective collection of adverse event data. The potential influence of confounding factors, such as adjustments to concomitant antiseizure medications, was not analyzed due to the sample size constraints. Future research should prioritize prospective designs with larger cohorts, systematic safety monitoring, and robust statistical control for confounders to validate these preliminary findings and precisely elucidate the temporal relationship between sleep improvement and seizure control.

## Conclusion

This study demonstrates for the first time that in children with SeLECTS treated with perampanel, the improvement in sleep quality and architecture precedes the significant reduction in seizure frequency and Spike-Wave Index. This suggests that PER, likely through its AMPA receptor antagonism, may preferentially stabilize the sleep-wake circuitry. Breaking the vicious cycle of "epilepsy–sleep disorder" provides a neural environmental foundation for subsequent seizure control. Future prospective, multicenter studies are warranted to validate sleep parameters as predictive biomarkers for PER efficacy.

## Supporting information

**S1 Data. Data.**

(XLS)

## Acknowledgments

The authors gratefully acknowledge the children and their families for their participation in this study. We also extend our thanks to the clinical team at the Department of Pediatric Neurology, Shaoxing People's Hospital, for their support.

## Author contributions

**Conceptualization:** Xiaoli Shao.

**Data curation:** Biao Jiang, Ninger Zhang.

**Formal analysis:** Biao Jiang.

**Investigation:** Biao Jiang, Ninger Zhang.

**Methodology:** Ninger Zhang.

**Project administration:** Xiaoli Shao.

**Resources:** Xiaoli Shao.

**Supervision:** Xiaoli Shao.

**Validation:** Ninger Zhang.

**Writing – original draft:** Biao Jiang.

**Writing – review & editing:** Xiaoli Shao.

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
