## [Decision Letter · Decision Letter 0]

17 Nov 2025

Dear Dr. Shao,

Thank you for submitting your manuscript to PLOS ONE. After careful consideration, we feel that it has merit but does not fully meet PLOS ONE’s publication criteria as it currently stands. Therefore, we invite you to submit a revised version of the manuscript that addresses the points raised during the review process.

**ACADEMIC EDITOR:**  **- our expert reviewers identified some major issues in the study design and results interpretation. Having in mind timeframes, please do make significant efforts to properly address all their comments. As soon as we receive your R1 it will be reviewed again.**

We look forward to receiving your revised manuscript.

Kind regards,

Prof. Dr. Dragan Hrncic, MD, MSc, MBE, PhD

Academic Editor

PLOS ONE

Journal Requirements:

“This work was supported by the following awards to X.-l.S.:

Grant number 2021ZYC-A23 from the Zhejiang Provincial Medical Association Clinical Research Fund

Grant number 2024ZF166 from the Zhejiang Provincial Traditional Chinese Medicine Health Technology Plan Project

Grant number 2023A14020 from the Shaoxing General Scientific Research Project

Grant numbers 2020KY327 and 2017KY660 from the Zhejiang Provincial Medical and Health Science and Technology Project

The sponsors or funders played no role in the study design, data collection and analysis, decision to publish, or preparation of the manuscript.”

Reviewers' comments:

Reviewer's Responses to Questions

**Comments to the Author**

1. Is the manuscript technically sound, and do the data support the conclusions?

Reviewer #1: Partly

Reviewer #2: Partly

Reviewer #3: Partly

2. Has the statistical analysis been performed appropriately and rigorously?

Reviewer #1: Yes

Reviewer #2: Yes

Reviewer #3: Yes

3. Have the authors made all data underlying the findings in their manuscript fully available?

Reviewer #1: Yes

Reviewer #2: Yes

Reviewer #3: Yes

4. Is the manuscript presented in an intelligible fashion and written in standard English?

Reviewer #1: Yes

Reviewer #2: Yes

Reviewer #3: Yes

Reviewer #1: This manuscript focuses on the "sleep-first effect" of perampanel (PER) in children with Self-Limited Epilepsy with Centrotemporal Spikes (SeLECTS). The topic holds clinical value, with a generally standardized study design, data collection and statistical methods that align with domain consensus, and a certain correlation between results and conclusions. However, there are shortcomings in the core argumentation, and the manuscript requires revision and improvement before further consideration for publication.

## Major Strengths

1. It is the first study to clearly identify the temporal sequence of "sleep improvement preceding seizure control" during PER treatment, filling the research gap regarding the temporal efficacy of PER in children with SeLECTS.

2. Reliable assessment tools were used (24-hour ambulatory EEG, Chinese version of the Children’s Sleep Habits Questionnaire [CSHQ]), and the study complies with ethical requirements and data standards, meeting the basic publication criteria of PLOS ONE.

## Key Limitations

1. The evidence supporting the "association between sleep improvement and reduced seizures" is insufficient. Only temporal correlation was presented, while quantitative analyses (e.g., correlation coefficients between sleep indicators and seizure-related indicators) and causal verification (e.g., additional effects of sleep interventions) are lacking.

2. There is a logical leap: "temporal sequence" is directly equated to "causal relationship," and the possibility that PER acts independently on sleep and seizure pathways was not excluded.

3. Confounding factors (e.g., adjustments to concomitant antiseizure medications [ASMs], age differences) were not analyzed via stratification, and safety data are missing, which undermines the reliability of the conclusions.

## Revision Suggestions

1. Supplement quantitative correlation analyses between sleep and seizure indicators, as well as multivariate regression analyses, to control for confounding factors.

2. Revise the conclusion statements to clearly emphasize an "association" rather than a "causal relationship."

3. Retrospectively extract adverse event data to improve the safety report.

Reviewer #2: Self-limited epilepsy wiith centrotemporal spikes is a benign epilepsy. The use of Perampanel is not justified in these patients. Also, the text mentions "Consistent with the self-limiting nature of SeLECTS, Perampanel (PER) was not utilized as a first-line therapy" but, there are patients who are on PER monotherapy.

Although the idea that Perampanel may be beneficial to sleep-first, seizure in follow-up, the study design is not sufficient to justify the hypothesis.

Reviewer #3: This manuscript examines the effects of perampanel in patients with SeLECTS, separating sleep-related outcomes from seizure and electrophysiological outcomes and considering their temporal evolution. The analytical methods are generally appropriate, but the interpretation is somewhat overstated. I believe the paper would be suitable for publication if the following points are addressed.

1. In the Methods, PER is described as add-on therapy to existing ASMs. However, Table 1 lists five patients as receiving “PER monotherapy” (0 concomitant ASDs). It is unclear at what time point concomitant ASDs were counted and how “monotherapy” is defined in this cohort. This should be clarified.

2. In the Discussion, the authors state that improvements in sleep preceded the reduction in seizures and SWI. However, all outcomes (sleep parameters, seizure frequency, and SWI) already show significant improvement at the 1-month time point. The data may be more compatible with a pattern in which sleep measures reach a plateau earlier, while seizures and SWI continue to improve between 1 and 2 months, rather than a clear “sleep first, seizures later” sequence. Thus, the temporal order in which sleep improvement leads to subsequent clinical and electrophysiological improvement is not clearly established. I recommend revising the description to reflect this.

3. The term “dual mechanism” is used, but not clearly defined. It is unclear whether this refers to two parallel effects (direct antiseizure and direct sleep effects) or to a causal chain in which sleep improvement secondarily drives seizure/EEG improvement. I suggest explicitly defining what is meant by “dual mechanism” in the manuscript.

4. If “dual mechanism” is intended to mean that PER-induced sleep improvement secondarily leads to electrophysiological and clinical improvement, this appears inconsistent with the data, which show significant changes in both sleep measures and seizure/SWI already at the 1-month time point. The results may be better interpreted as reflecting concurrent improvements in both domains rather than a clear causal sequence. The interpretation of the dual mechanism should therefore be adjusted accordingly.

5. The idea that sleep fragmentation lowers seizure threshold is broadly accepted and supported in some syndromes, especially idiopathic generalized epilepsies. However, strong evidence in other epilepsies, and particularly in SeLECTS, appears limited. The authors should clarify whether specific data exist for SeLECTS; if not, the claim should be presented as an extrapolation and the wording made more cautious.

6. Even if sleep measures improve earlier, or more strongly, than seizures and SWI, this does not prove that sleep improvement causes the subsequent seizure/EEG changes. One possible interpretation of the results is that PER, as a drug effect, directly deepens and consolidates sleep earlier, while its effect on epileptic activity simply emerges later. Alternatively, a reverse causal relationship is also plausible, in which partial improvements in seizures or SWI occurring in parallel lead to marked improvement in sleep. Given the single-arm, retrospective design without a control group, the causal link proposed by the authors should be treated as a hypothesis rather than an established mechanism, and alternative explanations should be more fully discussed.

**Do you want your identity to be public for this peer review?** For information about this choice, including consent withdrawal, please see our For information about this choice, including consent withdrawal, please see our Privacy Policy .

Reviewer #1: **Yes:** Mengyang WangMengyang Wang

Reviewer #2: No

Reviewer #3: **Yes:** Tohru OkanishiTohru Okanishi

---

## [Author Response · Author response to Decision Letter 1]

25 Dec 2025

Response to Reviewers

Manuscript ID: PONE-D-25-55018

Title: The Sleep-First Effect of Perampanel in Children with Self-Limited Epilepsy with Centrotemporal Spikes: A Temporal Efficacy Analysis in the Context of Complex Polytherapy

Dear Prof. Dr. Dragan Hrncic and Reviewers,

We sincerely thank you for your insightful comments and constructive suggestions on our manuscript. These comments have been invaluable in helping us to strengthen our study. We have carefully addressed all points raised during the review process. The changes in the manuscript have been highlighted using the track changes function. Our point-by-point responses are provided below.

The details of all funding sources for this study are provided in the funding statement within this cover letter.

Thank you for your note. We have now uploaded the minimal data set as a Supporting Information file with the submission, titled "S1_Data.xlsx".

Responses to Academic Editor

Comment: Our expert reviewers identified some major issues in the study design and results interpretation. Having in mind timeframes, please do make significant efforts to properly address all their comments.

Response: We deeply appreciate the editor's guidance. We have invested significant effort in a comprehensive revision to address all the concerns raised by the reviewers. Major revisions have been made to the Discussion, Methods, and Results sections to improve the clarity of our definitions, temper the interpretation of our findings, and enhance the overall rigor of the manuscript. We believe these changes have substantially improved the quality of our work.

Responses to Reviewer #1

Comment 1: The evidence supporting the "association between sleep improvement and reduced seizures" is insufficient. Only temporal correlation was presented... There is a logical leap: "temporal sequence" is directly equated to "causal relationship," and the possibility that PER acts independently on sleep and seizure pathways was not excluded.

Response: We thank the reviewer for this critical insight. We fully agree that a temporal sequence does not establish causality. We have thoroughly revised the Discussion section to address this point. Specifically, we have replaced definitive causal language with hypothetical and associative terms (e.g., “suggests,” “may help,” “could contribute to”). More importantly, we have added a new paragraph explicitly stating the limitation of our observational design in proving causation and acknowledging the parallel pathway possibility.( The relevant changes have been made in the Revised Manuscript with Track Changes on lines 179-189, 247-252.)

Comment 2: Confounding factors (e.g., adjustments to concomitant antiseizure medications [ASMs], age differences) were not analyzed via stratification, and safety data are missing, which undermines the reliability of the conclusions.

Response: We thank the reviewer for pointing this out.

Safety Data: We have now added a dedicated "Safety and Tolerability" subsection in the Results section, reporting the retrospective findings.

Confounding Factors: We acknowledge this as a valid limitation. Due to the relatively small sample size, a meaningful stratified analysis was not feasible. We have now explicitly mentioned this limitation in the Limitations paragraph of the Discussion.

Action: Please see the new "Safety and Tolerability" subsection in the Results and the updated Limitations section.( The relevant changes have been made in the Revised Manuscript with Track Changes on lines 137-144.)

Responses to Reviewer #2

Comment 1: Self-limited epilepsy with centrotemporal spikes is a benign epilepsy. The use of Perampanel is not justified in these patients.

Response: We understand the reviewer's concern. We have revised the Discussion to better articulate the clinical rationale. We emphasize that while SeLECTS is generally self-limited, a subset of children experiences clinically significant seizures and sleep disturbances that impair quality of life, justifying therapeutic intervention. Our cohort was specifically enrolled based on these active symptoms (seizure frequency ≥1/month and significant sleep disturbance).

Action: Please see the strengthened rationale in the Discussion sections.( The relevant changes have been made in the Revised Manuscript with Track Changes on lines 173-178.)

Comment 2: Also, the text mentions “Consistent with the self-limiting nature of SeLECTS, Perampanel (PER) was not utilized as a first-line therapy” but, there are patients who are on PER monotherapy.

Response: We sincerely apologize for this lack of clarity. We have clarified this point in the Methods section under “Treatment Protocol.” We now explain that the five patients on “PER monotherapy” had initially received PER as add-on therapy, and their prior medications were successfully tapered off before study enrollment due to excellent response. Thus, “monotherapy” refers to their status at the time of the study.

Action: Please see the clarified text in the Methods section.( The relevant changes have been made in the Revised Manuscript with Track Changes on lines 58-62.)

Responses to Reviewer #3

Comment 1: In the Methods, PER is described as add-on therapy to existing ASMs. However, Table 1 lists five patients as receiving “PER monotherapy”... This should be clarified.

Response: Please see our response and action to Reviewer #2, Comment 2. This has been clarified in the Methods section. (The relevant changes have been made in the Revised Manuscript with Track Changes on lines 58-62.)

Comment 2 & 4: In the Discussion, the authors state that improvements in sleep preceded the reduction in seizures and SWI. However, all outcomes... already show significant improvement at the 1-month time point... Thus, the temporal order... is not clearly established. I recommend revising the description... The results may be better interpreted as reflecting concurrent improvements in both domains rather than a clear causal sequence.

Response: This is a very astute observation. We have revised our description throughout the manuscript (Abstract, Results, Discussion) to more accurately reflect the data. We now describe the pattern as “sleep parameters improving and reaching a plateau earlier, while seizures and SWI showed continued improvement between 1 and 2 months,”rather than an absolute sequence.

Action: Please see the revised descriptions in the Abstract, Results, and Discussion.( The relevant changes have been made in the Revised Manuscript with Track Changes on lines 118-121, 134-136.)

Comment 3: The term “dual mechanism” is used, but not clearly defined.

Response: We thank the reviewer for this suggestion. We have now explicitly defined what we mean by “dual mechanism” in the Discussion section, presenting it as two potentially interconnected pathways: direct antiseizure and direct pro-sleep effects.

Action: Please see the defined term in the Discussion section.( The relevant changes have been made in the Revised Manuscript with Track Changes on lines 190-193.)

Comment 5: The idea that sleep fragmentation lowers seizure threshold... in SeLECTS, appears limited. The authors should clarify... the claim should be presented as an extrapolation and the wording made more cautious.

Response: We agree and have tempered our language. We now state that while this phenomenon is well-established in certain epilepsies like IGE, evidence in SeLECTS is still accumulating, and our results provide support for its existence in this syndrome.

Action: Please see the more cautious wording in the Discussion section.( The relevant changes have been made in the Revised Manuscript with Track Changes on lines 202-206.)

Once again, we extend our sincere gratitude to you for your time and invaluable comments, which have significantly strengthened our manuscript. We hope the revised version is now acceptable for publication in PLOS ONE.

Sincerely,

Xiao-li Shao, MS(Corresponding Author)

Pediatrics, Shaoxing People’s Hospital

The First Affiliated Hospital of Shaoxing University

Shaoxing, Zhejiang, 312000, China

Email: nuannuan717@126.com

On behalf of all co-authors:

Biao Jiang, Ning-er Zhang.

---

## [Decision Letter · Decision Letter 1]

4 Feb 2026

Dear Dr. Shao,

Thank you for submitting your manuscript to PLOS ONE. After careful consideration, we feel that it has merit but does not fully meet PLOS ONE’s publication criteria as it currently stands. Therefore, we invite you to submit a revised version of the manuscript that addresses the points raised during the review process.

We look forward to receiving your revised manuscript.

Kind regards,

Prof. Dr. Dragan Hrncic, MD, MSc, PhD

Academic Editor

PLOS One

Journal Requirements:

Additional Editor Comments:

please do make corrections as directed by reviewer comment. We are grateful to reviewers responding despite holiday season upon submission of R1 version,

Reviewers' comments:

Reviewer's Responses to Questions

**Comments to the Author**

Reviewer #1: All comments have been addressed

2. Is the manuscript technically sound, and do the data support the conclusions?

Reviewer #1: Yes

3. Has the statistical analysis been performed appropriately and rigorously?

Reviewer #1: Yes

4. Have the authors made all data underlying the findings in their manuscript fully available?

Reviewer #1: Yes

5. Is the manuscript presented in an intelligible fashion and written in standard English?

Reviewer #1: No

Reviewer #1: The table 2 safety data mentions serious adverse events, with a mix of English abbreviations and full terms, and the abbreviation "Aes" should be corrected to "AEs". It is suggested that only abbreviations be used in the table, and the full term of AEs be annotated below the table.

**Do you want your identity to be public for this peer review?** For information about this choice, including consent withdrawal, please see our For information about this choice, including consent withdrawal, please see our Privacy Policy .

Reviewer #1: **Yes:** Mengyang WangMengyang Wang

---

## [Author Response · Author response to Decision Letter 2]

23 Feb 2026

Dear Prof. Dr. Dragan Hrncic and Reviewers,

Thank you very much for your letter and for the reviewers’ constructive comments concerning our manuscript entitled "The Sleep-First Effect of Perampanel in Children with Self-Limited Epilepsy with Centrotemporal Spikes: A Temporal Efficacy Analysis in the Context of Complex Polytherapy" (Manuscript ID: PONE-D-25-55018R1). We appreciate the time and effort that you and the reviewers have dedicated to providing valuable feedback on our work. We have carefully considered all the comments and have revised the manuscript accordingly.

The modifications are highlighted in the revised manuscript using the "Track Changes" mode for your convenience. Below is our point-by-point response to the reviewer's comments.

Reviewer #1:

Comment: The table 2 safety data mentions serious adverse events, with a mix of English abbreviations and full terms, and the abbreviation "Aes" should be corrected to "AEs". It is suggested that only abbreviations be used in the table, and the full term of AEs be annotated below the table.

Response: Thank you for your careful observation and valuable suggestion. We completely agree with your assessment regarding the consistency and clarity of Table 2.

We have made the following revisions to Table 2 as requested:

Corrected the abbreviation: The erroneous term "Aes" has been corrected to "AEs" (Adverse Events) throughout the table.

Standardized terminology: To ensure consistency, we have revised the table to use only abbreviations for all adverse event terms.

Added a footnote: Below Table 2, we have added a clear footnote to define the abbreviations used. The footnote now reads: "AEs: Adverse Events; SAEs: Serious Adverse Events."

We believe these changes have significantly improved the clarity and professionalism of the data presentation in Table 2. We thank the reviewer for pointing out this oversight.

Once again, we extend our sincere gratitude to you for your time and invaluable comments, which have significantly strengthened our manuscript. We hope the revised version is now acceptable for publication in PLOS ONE.

Sincerely,

Xiao-li Shao, MS(Corresponding Author)

Pediatrics, Shaoxing People’s Hospital

The First Affiliated Hospital of Shaoxing University

Shaoxing, Zhejiang, 312000, China

Email: nuannuan717@126.com

On behalf of all co-authors:

Biao Jiang, Ning-er Zhang.

---

## [Editor Report · Decision Letter 2]

25 Feb 2026

The Sleep-First Effect of Perampanel in Children with Self-Limited Epilepsy with Centrotemporal Spikes: A Temporal Efficacy Analysis in the Context of Complex Polytherapy

PONE-D-25-55018R2

Dear Dr. Shao,

We’re pleased to inform you that your manuscript has been judged scientifically suitable for publication and will be formally accepted for publication once it meets all outstanding technical requirements.

Kind regards,

Prof. Dr. Dragan Hrncic, MD, PhD

Academic Editor

PLOS One
---

## [Editor Report · Acceptance letter]

PONE-D-25-55018R2

PLOS One

Dear Dr. Shao,

I'm pleased to inform you that your manuscript has been deemed suitable for publication in PLOS One. Congratulations! Your manuscript is now being handed over to our production team.

Kind regards,

on behalf of

Professor Dragan Hrncic

Academic Editor

PLOS One